# Development of a Model for Chemical Screening Based on Collateral Sensitivity to Target BTK C481S Mutant

**DOI:** 10.3390/cancers12040901

**Published:** 2020-04-07

**Authors:** Camille Libre, Ludovic Moro-Sibilot, Stéphane Giraud, Laetitia Martin, Els Verhoeyen, Caroline Costa, Amel Chebel, Nathalie Bissay, Gilles Salles, Laurent Genestier, Pierre Sujobert

**Affiliations:** 1Cancer Research Center of Lyon, INSERM U1052 UMR CNRS 5286, Equipe labellisée Ligue Contre le Cancer, Université de Lyon, 69008 Lyon, France; camillelibre59300@gmail.com (C.L.); pierre_sujobert@yahoo.fr (L.M.-S.); amel.chebel@univ-lyon1.fr (A.C.); nathalie.bissay@univ-lyon1.fr (N.B.); gilles.salles@chu-lyon.fr (G.S.); laurent.genestier@inserm.fr (L.G.); 2Center for Drug Discovery and Development (C3D), Fondation Synergie Lyon Cancer, Cancer Research Center of Lyon (CRCL, UMR INSERM 1052 CNRS 5286—Centre Léon Bérard), 69008 Lyon, France; Stephane.GIRAUD@lyon.unicancer.fr (S.G.); laetitia.martin@lyon.unicancer.fr (L.M.); 3CIRI—International Center for Infectiology Research, Inserm, U1111, Ecole Normale Supérieure de Lyon, Université Lyon 1, CNRS, 69364 UMR5308 Lyon, France; els.verhoeyen@ens-lyon.fr (E.V.); caroline.costa@ens-lyon.fr (C.C.); 4C3M, Université Côte d’Azur, INSERM, 06204 Nice, France

**Keywords:** BTK, collateral sensitivity, chemical screen, ibrutinib resistance

## Abstract

Targeted therapies have improved the outcome of cancer, but their efficacy is intrinsically limited by the emergence of subclones with a mutation in the gene encoding the target protein. A few examples of collateral sensitivity have demonstrated that the conformational changes induced by these mutations can create unexpected sensitivity to other kinase inhibitors, but whether this concept can be generalized is unknown. Here is described the development of a model to screen a library of kinase inhibitors for collateral sensitivity drugs active on the Bruton Tyrosine Kinase (BTK) protein with the ibrutinib resistance mutation C481S. First, we demonstrate that overexpression of the constitutively active mutant of BTK harboring the E41K mutation in Ba/F3 cells creates an oncogenic addiction to BTK. Then, we have exploited this phenotype to perform a screen of a kinase inhibitor library on cells with or without the ibrutinib resistance mutation. The BTK inhibitors showed the expected sensitivity profile, but none of the drugs tested had a specific activity against the C481S mutant of BTK, suggesting that extending the collateral sensitivity paradigm to all kinases targeted by cancer therapy might not be trivial.

## 1. Introduction

With the development of targeted therapies, the emergence of subclones with a resistance mutation has become a major concern of cancer treatment. The collateral sensitivity concept is a promising strategy to overcome this problem: the conformational changes induced by a mutation conferring resistance to a given compound can create a specific sensitivity to another compound [1]. Following this concept, it is appealing to develop repurposing screens to identify molecules with a high level of specificity against the mutated clone, which can be easily transferred to the clinics. The best example of collateral sensitivity described so far is axitinib, which is able to inhibit BCR-ABL1 harboring the T315I gatekeeper mutation of ABL1, but is ineffective against wild type BCR-ABL1 [2]. Another convincing example has been reported with the L1198F mutation of ALK, which confers resistance to lorlatinib but restores sensitivity to crizotinib in ALK rearranged lung cancers [3]. Whether the concept of collateral sensitivity is restricted to these few examples or could be useful in other situations of kinase inhibitor resistance is still unknown. Here we describe the development of a cellular model optimized for a chemical screen designed to discover collateral sensitivity drugs to overcome resistance to the BTK inhibitor ibrutinib.

B cell receptor (BCR) signaling has been recognized as critical in B cell lymphomagenesis, leading to the activation of oncogenic pathways such as NF-kB, PI3K/AKT and MAP-Kinases [4]. Bruton tyrosine kinase (BTK), assuring one of the first steps in BCR signaling, has become a therapeutic target in lymphoid malignancies. Ibrutinib is the first in class irreversible BTK inhibitor which binds covalently to the cysteine residue 481. Ibrutinib has demonstrated clinical activity in various lymphoid malignancies such as chronic lymphocytic leukemia (CLL), mantle cell lymphoma, marginal zone lymphoma, and Waldenström macroglobulinemia [5].

In CLL, about a quarter of the patients treated with ibrutinib show disease progression during the first four years [6]. Progressions are mostly driven by the emergence of subclones harboring the BTK cysteine 481 mutation (p.C481S substitution being most frequent) [7]. This mutation prevents the covalent binding of ibrutinib, and confers an increased fitness under the selective pressure of ibrutinib [8]. Of note, some patients have multiple subclones with different mutations in the BTK pathway, including mutations in PLCγ2, which is the direct target of BTK. Given the poor reported outcome of patients after ibrutinib discontinuation, at least in the pre-venetoclax era [9], new therapeutic options for patients harboring the BTK C481S mutation represent an unmet need.

Here, we describe the development of a cellular model to screen large chemical libraries for the identification of compounds with collateral sensitivity against BTK-C481S cells.

## 2. Results

### 2.1. Establishment of a Cell Line for Collateral Sensitivity Screen

In order to screen large libraries of compounds for collateral sensitivity against the BTK-C481S mutant, we engineered a cell line whose survival is absolutely dependent on the activity of BTK (either WT or C481S). The Ba/F3 cell line is a murine pro-B cell line, which requires murine IL3 in the culture medium to survive. This cell line has been largely used as a model to demonstrate the transforming potential of oncogenes, which are able to confer IL3 independence to the Ba/F3 cell line [10]. We overexpressed in Ba/F3 cells either human BTK^WT^ (in an IRES-mCherry lentiviral backbone) or human BTK^E41K^ with the E41K mutation in the pleckstrin-homology domain, conferring constitutive activation of the kinase (in an IRES-GFP lentiviral backbone) [11]. After IL3 starvation, we observed an activation of the BTK and PI3K pathway in Ba/F3 cells overexpressing BTK^E41K^. But not in Ba/F3 cells overexpressing BTK^WT^ (Figure 1A and Appendix A). Accordingly, we observed that the overexpression of BTK^E41K^ conferred IL3 independency to Ba/F3 cells, whereas the overexpression of BTK^WT^ did not (Figure 1B). To confirm this observation, we performed a competitive assay of BTK^WT^ (mCherry positive) versus BTK^E41K^ (GFP positive) expressing Ba/F3. Whereas we did not observe any modification of the clonal composition when the cells were cultured with IL3 (49% GFP positive cells), we observed a positive selection of BTK^E41K^ cells after IL3 deprivation (93% GFP positive cells) (Figure 1C). Of note, we observed that Ba/F3 cells expressing BTK^E41K^ became highly sensitive to ibrutinib in the absence of IL3, but not in the IL3-containing medium, confirming that the expression of BTK^E41K^ created an oncogenic addiction in Ba/F3 cells in the absence of IL3 (Figure 1D).

Having established a cell line whose survival is totally dependent on human BTK kinase activity in the absence of IL3, we further refined this cell model to establish a collateral sensitivity screen. Accordingly, we overexpressed human BTK with both the activating mutation E41K and the ibrutinib resistance mutation C481S (BTK^E41K-C481S^). As expected, the C481S mutation conferred resistance to ibrutinib (IC50 10µM vs. 100 nM for BTK^E41K^) (Figure 1D) and abolished the effects of ibrutinib on BTK signaling (Figure 1E).

### 2.2. Collateral Sensitivity Screen

In order to maximize our chances to identify compounds able to exploit the collateral sensitivity created by the C481S mutation, we have chosen to test a library of 590 kinase inhibitors (Medchemexpress, Monmouth Junction, NJ, USA). The output of the screen was the cell viability (as compared to vehicle (DMSO)), estimated by automated image analysis of GFP positive cells after a 48 h culture using the Operetta device.

The screen compared the viability of Ba/F3 overexpressing either BTK^E41K^ or BTK^E41K-C481S^ in IL3 depleted medium, in the presence of drugs at 100 nM (screen 1) or 1 μM (screen 2) or equivalent quantities of DMSO. We calculated a differential score as follow:(1)$$d={\mathrm{viability}}_{E41K-C481S}-{\mathrm{viability}}_{E41K}$$

For each drug, we plotted the value of d at 100 nM and 1 μM for graphical representation of the screens (Figure 2A, and Appendix A). Accordingly, the drugs with preferential activity against BTK^E41K^ should fall in the upper right quadrant, whereas the drugs with preferential activity against BTK^E41K-C481S^ (i.e., with collateral sensitivity) should fall in the lower left quadrant.

The analysis of the different BTK inhibitors contained in the library confirmed the accuracy of our screening approach, because most of them were in the expected upper right quadrant (Figure 2B, red dots, and Appendix A), except for CGI-1746 [12] which showed differential effects only at 1 μM and CNX-774 which had no differential effect at both doses. Of note, the racemate forms of ibrutinib (mixture of dextrogyre and levogyre) did not fall into this quadrant (Figure 2B, blue dot). When analyzing the different classes of compounds according to their described targets, we found an unexpected trend for preferential effect of PI3K inhibitors on BTK^E41K^ expressing cells (Figure 2C and Appendix A). An independent experiment confirmed that the C481S mutation conferred resistance to idelalisib (a p110δ inhibitor) in the Ba/F3 model (Appendix A). To investigate the observation suggesting that the BTK C481S mutation could confer a cross resistance to PI3K inhibitors further, we overexpressed BTK^WT^ or BTK^C481S^ in a human diffuse large B-cell lymphoma cell line sensitive to ibrutinib (TMD8). We confirmed that BTK^C481S^ confers ibrutinib resistance in this cell line, but we did not detect any differential effect with idelalisib treatment (Appendix A).

Finally, there was no drug with a clear specific activity against BTKE41K-C481S in this kinase inhibitor library. We selected 11 drugs (BMS536924, dinaciclib, alisertib, LCK inhibitor, MK1775, LY2603618, cerdulatinib, AZD6738, THZ1, AZD1152, purvanalol) which showed more pronounced activity on BTKE41K-C481S at least at one concentration (100 nM or 1 μM), and tested them on a large dose range in independent experiments. The insulin receptor IGF-1R inhibitor BMS536924 was the only drug with a confirmed specific activity against Ba/F3 overexpressing BTKE41K-C481S over a large range of doses, but this was not confirmed in the lymphoma cell line TMD8 (Appendix A).

## 3. Discussion

We have developed a cellular model allowing the screening of a large number of compounds for their capacity to inhibit BTK with or without the C481S mutation. The use of the Ba/F3 cell line is a strategy which could be applied with other mutated oncogenes able to transform these cells, such as FLT3-ITD [13] BRAF V600E [14] or PIK3CA mutants [15]. Indeed, the expression of an oncogenic driver creates an oncogenic addiction in Ba/F3 cells after IL3 deprivation, which is an optimal phenotype for screening [10]. This strategy can also be used for therapeutic targets that are not mutated in cancer, such as BTK. However, in these situations, wild type proteins might be unable to transform the Ba/F3, and constitutively activated mutants might be required, which introduces a potential experimental bias.

Our data show that this screening strategy has a strong discriminating power, because BTK inhibitors have the expected profile of efficacy. However, some hits should be interpreted with caution (such as the PI3K inhibitors), because the basis of their preferential effects against BTK^E41K^ as compared to BTK^E41K-C481S^ in the Ba/F3 cell line is unclear. Of note, we should keep in mind that we introduced the E41K activating mutation of BTK to confer IL3 independence to the Ba/F3, and we cannot exclude that this additional mutation might explain the preferential effect of PI3K inhibitors on BTK^E41K^ expressing Ba/F3. Accordingly, the differential sensitivity to PI3K inhibitors was not confirmed in lymphoma cell lines overexpressing BTK^WT^ or BTK^C481S^ without the E41K mutation.

## 4. Materials and Methods

### 4.1. Cell Lines and Reagents

The cell lines were cultivated in RPMI-1640 media (Thermo Fisher Scientific, Waltham, MA, USA) supplemented with 100 U/mL penicillin, 100 mg/mL streptomycin, and 10% fetal calf serum. Ba/F3 were seeded at 5.105/mL in RPMI 1640 supplemented or not with murine IL3 (10 ng/mL, PeproTech, Rocky Hill, NJ, USA). The BTK inhibitor ibrutinib was purchased from Selleck Chemicals (Houston, TX, USA) and the kinase inhibitor library was purchased from MedChemExpress (Monmouth Junction, NJ, USA, cat #HY-L009).

### 4.2. Retroviral Transduction Experiments

We used murine retroviral vectors to express human BTK^WT^, BTK^E41K^, BTK^C481S^, or BTK^E41K-C481S^ in Ba/F3 and TMD8 cells. A murine retroviral vector, pMSCV-IRES, allowing co-expression of the human BTK gene and mCherry (Addgene ref #52114) or GFP (Addgene ref #20672) via an IRES element was used for transduction of Ba/F3 cells (Addgene, Cambridge, MA, USA). A lentiviral vector, pHIV-SFFV-MSC [16] allowing co-expression of BTK and mCherry or GFP via an IRES element was used for human TMD8 cells. Lentiviral vectors were produced as previously described [10]. Briefly, self-inactivating HIV-1–derived vectors were generated by transient transfection of 293T cells. For co-display of the different H and F measles glycoproteins, 3 μg of each envelope plasmid together with 8.6 μg of HIV-gagpol construct and 8.6 μg of lentiviral vector encoding plasmid were co-transfected. Viral supernatant was harvested 48 h after transfection. Low-speed concentration of the vectors was performed by overnight centrifugation of the viral supernatant at 3000 g at 4 °C.

The murine retroviral vectors were produced following the same protocol using 3 μg VSV-G envelope encoding plasmid, 8.6 μg of MLV-gagpol and 8.6 μg of vector or MSCV vector.

### 4.3. Screening Assay

For compound screening, Ba/F3 stably expressing BTK with the activating mutation E41K (BTK^E41K^) and Ba/F3 cells expressing BTK with both the activating mutation and the ibrutinib resistance mutation C481S on the same allele (BTK^E41K-C481S^) were seeded in 384-well plates at a density of 10,000 cells in 20 µL/well. The MedChemExpress library of 590 kinase inhibitors (kinase inhibitors in late development or already on the market) was then added to a final dose of 1µM or 100 nM (duplicates were performed, Appendix A). After 48 h of treatment, the cell numbers were quantified using the Perkin Elmer Operetta HCS system. Images were acquired using the EGFP channel (Ex 460–490 nm; Em 500–550; 5 fields per well, at objective 10×). Parameters of area, roundness and intensity were used to detect the cells, to perform segmentation and to calculate the number of living cells. Ibrutinib and Pentamidine were used as positive controls of cell death induction. The percentage of cell death was calculated using vehicle (0.5% DMSO) treated cells set to 100% of viability.

## 5. Conclusions

The absence of collateral sensitivity hits in this screen is disappointing from a clinical point of view. However, this result is important, because it suggests that the concept of collateral sensitivity is probably not so easily extendable to all kinases targeted by cancer therapy. Of course, other screens with different targets and different libraries will be necessary to comprehensively assess to what extent the concept of collateral sensitivity is useful to repurposing drugs for patients with drug resistance mutations. Regarding BTK C481S, the platform presented is a valuable tool for exploratory drug screening.

## Figures and Tables

**Figure 1 cancers-12-00901-f001:**
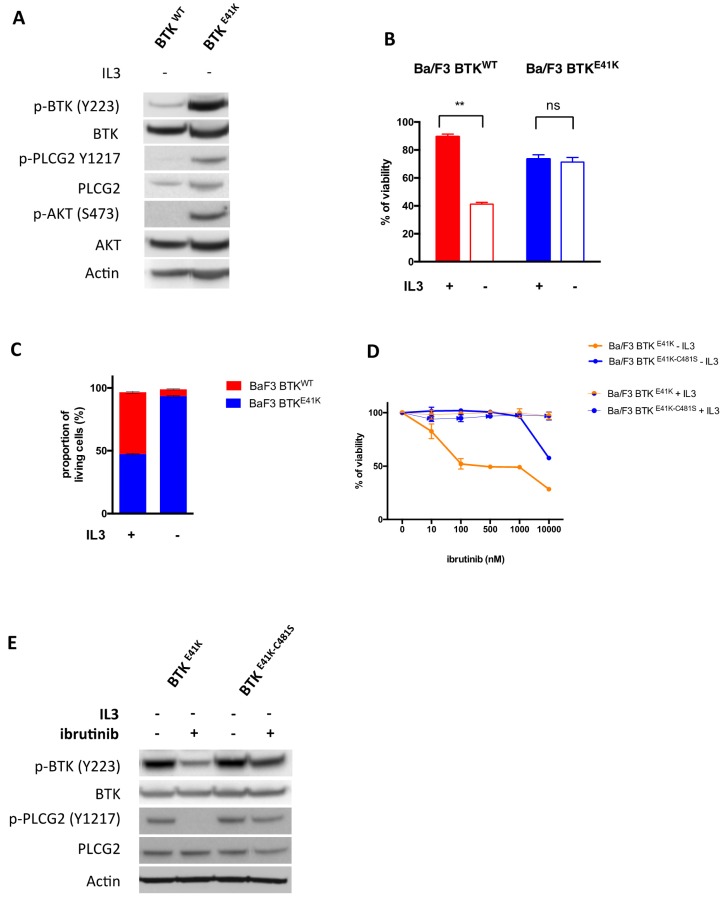
Engineering of a cell line to establish a collateral sensitivity screen. (**A**) Immunoblotting of Ba/F3 cells overexpressing either BTK^WT^ or BTK^E41K^ after a 4 h interleukin 3 (IL3) deprivation. (**B**) Proliferation assay of Ba/F3 cells overexpressing either BTK^WT^ or BTK^E41K^ with or without IL3. (**C**) Competitive assay of Ba/F3 cells overexpressing BTK^WT^ (mCherry positive) or BTK^E41K^ (GFP positive). The proportion of each cell type was measured after a 3 days culture with or without IL3. (**D**) Relative survival of ibrutinib treated Ba/F3 cells (as compared to vehicle treated cells) overexpressing either BTK^E41K-C481S^ or BTK^E41K^ in the presence (thin lines) or in the absence of IL3 (thick lines). (**E**) Immunoblotting of Ba/F3 cells overexpressing either BTK^E41K^ or BTK^E41K-C481S^ after a 4 h IL3 deprivation with or without 100 nM ibrutinib. ** *p* < 0.01; NS: Non significant. A t-test was used to assess the statistical significance of the results.

**Figure 2 cancers-12-00901-f002:**
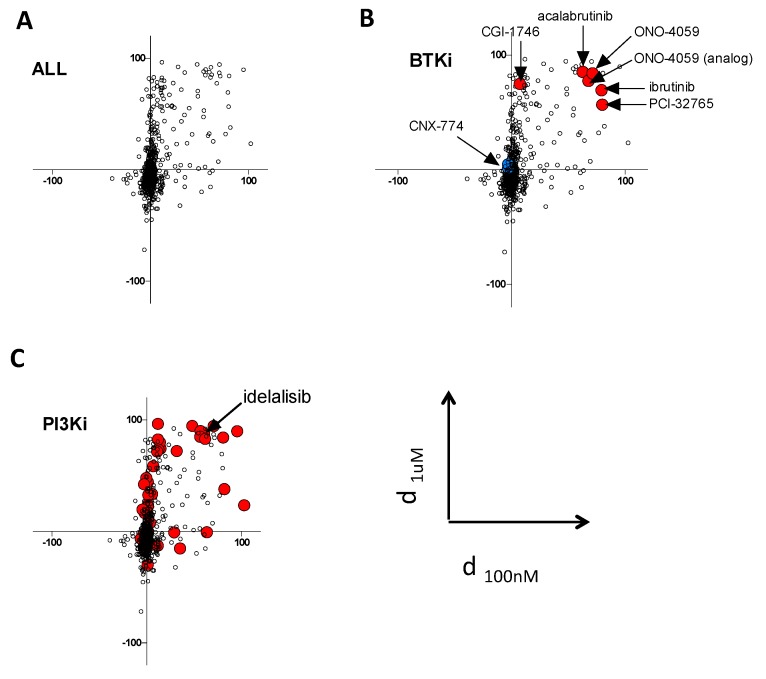
Collateral sensitivity screen. (**A**) The plot represents the differential effects of the compounds from the kinase inhibitors library on cellular viability in Ba/F3 cells expressing either BTK^E41K^ or BTK^E41K-C481S^. Each experimental condition (mean of duplicates) is plotted according to the differential effect at 100 nM (on the X axis) and 1 µM (on the Y axis); accordingly, compounds with selective efficacy against BTK^E41K^ fall in the right upper quadrant, and those with selective efficacy against BTK^E41K-C481S^ fall in the lower left quadrant. (**B**) BTK inhibitors are highlighted as large colored dots. (**C**) PI3K inhibitors are highlighted as large red dots. Idelalisib, which was further studied in another cell line, is highlighted.

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
