# Peer review of "Development of a Model for Chemical Screening Based on Collateral Sensitivity to Target BTK C481S Mutant"

_cancers, 2020, doi:10.3390/cancers12040901_

Round 1

Reviewer 1 Report

Manuscript describes development of a model for chemical screening of compounds against BTK C481S mutant. The aim of the study was to overcome ibrutinib resistance mutation C481S due to promoting oncogenic addiction to BTK by E41K overexpression. Double mutated BTKE41K-C481S, BTKwt and BTKC481S mouse Ba/F3 cells were established, as well as BTKwt and BTKC481S in human TDK8 cells. The study was based on the assumption that possible collateral sensitivity will reveal BTK-C481S inhibitors among the library compounds.

Obtained results did not support this idea, narrowing its applicability to certain cellular context and to rearranged or overexpressed proteins, maybe. Screening of a large number of potential anti-BTK compounds detected one single drug against Ba/F3 overexpressing BTKE41K-C481S which was not effective in human TMD8. Together, presented data evident against "collateral sensitivity" approach as a useful strategy for development of anti-BTK inhibitors.

Remarks:

1) Fig.1D: is a difference between presented survival curves significant?

2) Fig.2 A-C: "?" should be avoid; X and Y axis should be signed and explained;

Figure legend to Fig.2, line 139: 1 µM (on the Y axis): µ is missed.

Reviewer 2 Report

The paper by Libre et al. is devoted to development of a screening assay for discovery of drugs potentially effective after development of ibrutinib resistance. The manuscript contains serious flaws and I would not recommend publishing it in its present form.

The manuscript contains lots of misprints and grammar mistakes and requires extensive polishing of English. Some examples:

Lines 59, 76: looks like a special symbol is missing in “PLC 2”

“We have developed an in cellulo platform” – what does it mean?

“We overexpressed in Ba/F3 cells either human BTK WT”

BTKE41K – this is hard read, E41K should be in a subscript or superscript

Other issues:

IL3+ control experiments should be shown on Fig.1A and Fig.1E.

How many replicates were performed for viability experiments (Fig. 1B and 1D, Fig.S1)? Why there are no error bars on these panels? Such experiments should be performed at least in triplicates.

Column names in supplementary tables do not provide enough information about the content of the tables.

How do the authors explain outliers among BTK-inhibitors on Fig. 1B.

“Having noticed that 12 drugs seemed to be preferentially active on BTKE41K-C481S at at least one concentration, we tested them on a large dose range in independent experiments. The insulin receptor IGF-1R inhibitor BMS536924 was the only drug with a confirmed specific activity against Ba/F3 overexpressing BTKE41K-C481S over a large range of doses, but this was not confirmed in the lymphoma cell line TMD8” – what are these drugs? What was the criteria for selection of exactly 12 drugs? Why the authors do not show the results on another 11 drugs?

Moreover, from the manuscript text it is unclear whether proposed approach is first of its kind, or if not, what is the difference from the existing approaches.

Finally, the title is misleading: the authors aimed on developing a model for screening collateral sensitivity. However, implementation of the approach did not result  in any potential drug candidate. Isn’t it possible that the proposed approach is not suitable to perform such screening?

Round 2

Reviewer 2 Report

The authors improved the manuscript. However, I still have the following points, which should be addressed:

“For the Western blot presented in Fig 1A, we prefer to show only the conditions where the screen is done, i.e without IL3.” Why so? I believe showing IL3+ will be important, because it will indicate whether BTK, AKT and PLCG2 phosphorylation is dependent on IL3 presence.

 “When analyzing the different classes of compounds according to their described targets, we found an unexpected preferential effect on BTKE41K for PI3K inhibitors” – this analysis should be formalized, and the details should be provided. How did the authors measure “preferential effect”? Was this done manually? Information regarding other drug classes should also be provided. How many classes were analyzed and how many drugs did they contain?

 “We selected the drugs having the highest differential effect at 100nM or 1μM. The list of these drugs is now provided in the text.” – this looks like a formal criterion for selection of the 12 drugs. I think the authors should rephrase “Having noticed that 12 drugs seemed to be preferentially active…” accordingly.

Fig.2 and both supplementary figures should be modified: “?” symbols should be replaced with the appropriate symbols.

“BTKWT” – WT is in superscript in some cases, and in capital letters in the others. This should be unified across the manuscript.

Finally, I do believe that the manuscript would benefit significantly if the English language would be edited by a native speaker:

“We have developed an cellular model” – should be “a cellular model”

“ie. with collateral sensitivity” – should be “i.e.” and so on across the manuscript.

Round 3

Reviewer 2 Report

“At the biochemical level (figure 1A), we observed that contrary to Ba/F3 cells overexpressing BTKWT, Ba/F3 cells overexpressing BTKE41K were able to maintain a strong level of phosphorylation of BTK and its downstream kinase PLCγ2 in the absence of IL3, together with activation of the PI3K pathway.”

- as I see from the western blot analysis provided by the authors in their cover letter, this statement is misleading. BTKwt overexpressing cells maintain phosphorylation of BTK, AKT and PLCg after IL3 deprivation with minor changes. It looks like constitutive activation of BTK leads to an increased proliferative activity, that is seen in Fig. 1C. However, the growth of Ba/F3 BTKwt cells may be inhibited in the absence of IL3 by the other mechanisms. The entire fragment in the manuscript should be changed accordingly and the Western blot with phosphorylation kinetics should be added at least to supplementary materials together with quantification of signal intensity. In addition, total PLCg level should be shown on that Western blot.

“Here is an analysis of the screen at the two doses, when the drugs are grouped according to their target. Obviously only the BTK and PI3K inhibitors (and maybe EGFR inhibitors, but only at 1 uM) have a preferential action against BTK WT. We do not think that this figure has to be integrated in the manuscript.”

- these figures are important and could be integrated into supplementary materials together with detailed explanation. It is unclear why two panels of the figure have different number of compound classes and why +/- 50 cut-off was chosen. I do not agree that preferential action is obvious and insist on introducing the formal criteria for this analysis. If the result is obvious for the authors, there should be no problem for doing that. Of note: the difference in Fig.1B is also “obvious”, however the authors performed an unknown test (should also be clarified in the manuscript text) to calculate p-value proving that.

I am also disappointed that after two rounds of revision where I was pointing at different typos and oversights they  remain in the manuscript:

Line 151: “at at least one concentration”

Line 200: “via and IRES element”

I would like to ask the authors not only to correct these typos but rather to carefully revise entire manuscript before submitting it further.

Round 4

Reviewer 2 Report

The authors significantly improved the manuscript. I have two minor points:

1) I think that graphical representation of the class effect provided in the cover letter should also be incorporated into supplementary materials.

2) There are "?" signs on Fig. S1 and the caption is "IL3 depriva,on ,me", which should be corrected.

Author Response

We warmly thak the reviewer for his/her positive comment. We have done the minor modifications required, and included the supplemental figure required.

Regarding the "?" sign and the problems on figure s1, we believe that this is due to problems with the PDF conversion by the Editor. In the attached PDF version of the paper, these problems are resolved.